# Recovering from Misbehaviors in Coding Agents

## Abstract

Autonomous coding agents, powered by large language models (LLMs), are increasingly being adopted in the software industry to automate complex engineering tasks. However, these agents are prone to a wide range of misbehaviors, such as deviating from the user's instructions, getting stuck in repetitive loops, or failing to use tools correctly. These failures disrupt the development workflow and often require resource-intensive manual intervention. In this paper, we present a system for automatically recovering from agentic misbehaviors at scale. We first introduce a taxonomy of misbehaviors grounded in an analysis of production traffic, identifying three primary categories: *Specification Drift*, *Reasoning Problems*, and *Tool Call Failures*, which we find occur in about 30% of all agent trajectories.

To address these issues, we developed a lightweight, asynchronous self-intervention system that observes agent trajectories and provides targeted course-correction guidance to nudge the agent back to a productive path. We evaluated our system on over 10,000 real-world agent trajectories and found that it successfully resolves 90% of the misbehaviors that require a single intervention. Furthermore, a live A/B test in our production environment demonstrated that our system leads to a statistically significant reduction in *Tool Call Failures*, *Tokens per Session* and *Engineer Interventions per Session*. We present our experience designing and deploying this system, offering insights into the challenges of building resilient agentic systems at scale.

## 1 Introduction

*Coding Agents* Coding agents have gained popularity in the past year, and have been aggressively adopted in the industry, including at large IT companies. A *coding agent* is an autonomous system that utilizes a large language model (LLM) to perform software engineering tasks. Given a user query in natural language representing a high level objective and some context, often including source code files of a repository, the agent interacts with a software development environment to achieve its goal. This process typically involves iterative cycles of reasoning and action, where the agent selects from a set of available tools—such as reading files, generating code, or invoking a compiler—to make progress.

While agents have been reported to increase productivity, user experience when using agents is not uniform. Users still have to manually correct and steer the agent towards desired outcomes. There are multiple ways in which agents can fail to perform well. They may get stuck until a user comes in and steers them, or they may head towards incorrect paths, e.g. going off and modifying unintended files. They may also consume more steps than necessary. Users perceive as success when their request gets fulfilled correctly, efficiently, and as autonomously as possible. Agent misbehaviors have become an active area of study[3, 5] in its own right.

*Self intervention* In this work, we talk about the ways in which agents misbehave, and more importantly, also automated mechanisms to nudge these agents towards self recovery from these misbehaviors. The notion of agent "behavior" is typically an *execution trajectory*, which is a sequence of steps. Each step comprises the agent's internal rationale (i.e., the reasoning behind its next action), the action itself (e.g., a tool invocation), and the resulting observation from the environment. Analyzing these trajectories provides fine-grained insight into the agent's decision-making process and enables the identification of misbehavior patterns.

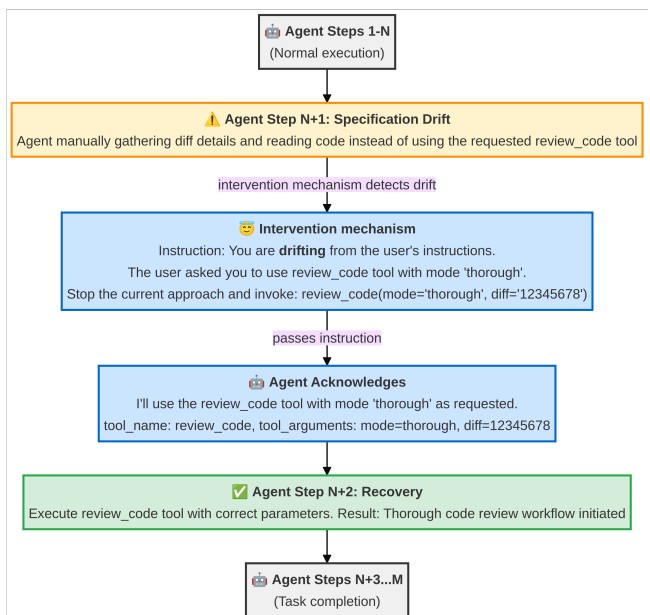

Figure 1: Self-intervention mechanism addressing specification drift. The agent deviates from user instructions, the intervention agent detects this drift, redirects the agent to follow the explicit instructions, and the agent acknowledges and executes the correct tool invocation, successfully completing the task.

Figure 1 illustrates a trajectory with a particular kind of misbehavior. The coding agent deviated from the user's explicit instructions by manually gathering diff details and reading code instead of using the requested `review_code` tool with mode `thorough`.

To mitigate such a misbehavior, our proposal is to use runtime periodic **intervention** to nudge an agent out of the misbehavior.

Continuing with Figure 1, the intervention mechanism that runs concurrently with the main coding agent—detected this drift from the specified instructions, identified the deviation, and provided corrected instructions to invoke the proper tool with the correct parameters. Upon receiving this course-correction, the main agent acknowledged the instruction, executed the `review_code` tool as originally requested, and successfully completed the task.

This is of course not the only kind of agent misbehavior: in this work we address three additional kinds of misbehaviors and their interventions.

*Production Setting* We have tested our interventions in a real production environment that supports thousands of developers who interact with LLM-powered coding assistant via a Visual Studio Code extension. These coding assistants support three primary use cases: program comprehension (e.g., code explanation and documentation), code generation (e.g., generating new code artifacts), and code review (e.g., bugs and style issues). The platform integrates numerous Model Context Protocol (MCP) tools that provide agents with access to proprietary development infrastructure. MCP tools serve as interfaces between the AI agents and internal resources, including version control systems, internal databases, build systems, and other development tooling. This architecture enables agents to perform context-aware operations within internal software ecosystem.

*Overview of Results* We prototyped our self intervention in the context of trajectories generated from the agentic coding IDE mentioned above. Self-intervention shows strong recovery rates across all misbehavior types: conversations with single-intervention achieving 90%, and conversations requiring multiple interventions achieving about 80% recovery. Moreover, we observed stat-sig reductions in tool call failures, token usage per session and engineer interventions per session during online A/B testing on production traffic.

## 2 Misbehaviors in Software Engineering Agents

Software engineering agents increasingly operate within complex, tool-rich development environments and exhibit characteristic failure modes during task execution. Prior work proposes taxonomy-driven classifications of such failures. For example, Deshpande et al. [9] present a formal taxonomy spanning reasoning, execution, and planning errors; Majgaonkar et al. [18] derive an empirically grounded taxonomy from real-world GitHub issues involving coding agents; and Gandhi et al. [12] introduce a domain-general taxonomy validated in software engineering settings.

Building on these foundations, we adopt a high-level taxonomy but re-operationalize the categories to fit an enterprise context with proprietary languages, org-specific frameworks, and heterogeneous legacy systems. We find that a bottom-up construction—grounded in production trajectories and developer feedback—is necessary to ensure construct validity and operational utility. Our goal is to quantify the prevalence of failure modes in day-to-day use and to surface actionable error classes that inform the design of runtime interventions, verification hooks, and agent tooling for large-scale industrial code bases.

### 2.1 Common Misbehaviors in Our Setting

Our IDE is instrumented so users can offer explicit feedback in the form of "Like" or "Dislike" to the coding agent. We perform a comprehensive review of disliked trajectories to uncover the most prevalent issues affecting real users. Furthermore, manual inspection of these disliked trajectories gives us a strong understanding of why users disliked a particular trajectory and what actions could be taken to recover from such issues. After studying hundreds of trajectories, we found that misbehaviors can be categorized into three main classes: Specification Drift (SD), Reasoning Problems (RP), Tool Call Failures (TCF).

*2.1.1 Specification Drift (SD)* This category of misbehavior captures instances where an agent diverges from the task specified by the user. Any deviation from the original user requirements is considered specification drift. We identify two primary subcategories:

*Did Not Follow Instructions (DNF)* This subcategory describes situations in which the agent fails to strictly adhere to the user's explicit instructions and intent. This includes cases where the agent ignores user constraints, provides solutions that are only tangentially related to the request, fails to incorporate user feedback, omits key details or context, or otherwise exceeds the desired scope through over-explanation or excessive editing.

*Unrequested Changes (UC)* This subcategory pertains to scenarios where the agent makes modifications that were not requested by the user or edits files unrelated to the user's instructions. This includes instances where the agent alters content outside the scope of the user's request, makes changes that do not align with the user's intent, or requires the user to intervene in order to revert these unrequested modifications.

*2.1.2 Reasoning Problems* This category of misbehaviors captures problems associated with agent's reasoning and thinking. Often, these issues impact the agent's ability to make meaningful progress towards task completion.

*Infinite Loops* Infinite loops are failure patterns in coding agents where the agent becomes stuck in a cycle of repetitive actions or reasoning, making little to no progress on the assigned task. This often manifests as repeated tool calls, unsuccessful attempts to fix self-introduced errors (like syntax or lint errors), or endless edits to resolve merge conflicts. Key indicators include the agent invoking the same or similar tool calls three or more times in a row, repeated code edits to the same file or engaging in verbose reasoning without advancing toward a solution.

*2.1.3 Tool Call Failures* This category refers to instances where the agent *repeatedly* fails to interact with tools due to its own errors or unresponsiveness. It encompasses situations in which the agent issues malformed, invalid, or incorrect parameters during tool invocation and does not correct these mistakes. Examples include providing wrong or invalid arguments, attempting to invoke non-existent tools, or omitting required parameters. Additionally, this category covers situations where the agent ignores tool invocation failures and fails to adjust its strategy in response.

### 2.2 Methodology for Misbehavior Prevalence Calculation

We validate that these misbehavior categories are prevalent outside of disliked set and measure their occurrence in production traffic using classifiers.

*2.2.1 Classifiers for prevalence tracking* We calibrate LLM-based classifiers for each misbehavior category. The LLM acts as a binary classifier over trajectory input which includes the conversation history until a given step. We evaluated a variety of frontier models for classification including Claude Sonnet models (4, 4.5), Claude Haiku 4.5, GPT-4o, GPT 5.1 and Gemini 2.5 Pro across individual

| Misbehavior Category | Trajectories Detected | Prevalence |
|---|---|---|
| Loops | 2232 | 5.21% |
| DNF | 6827 | 15.95% |
| UC | 2833 | 6.62% |
| Tool Call Failure | 6001 | 14.02% |
| **Total Misbehavior Categories** | **12,499** | **29.2%** |

Table 1: Prevalence of detected misbehavior categories in 42,807k trajectories sampled from production traffic over a period of five weeks.

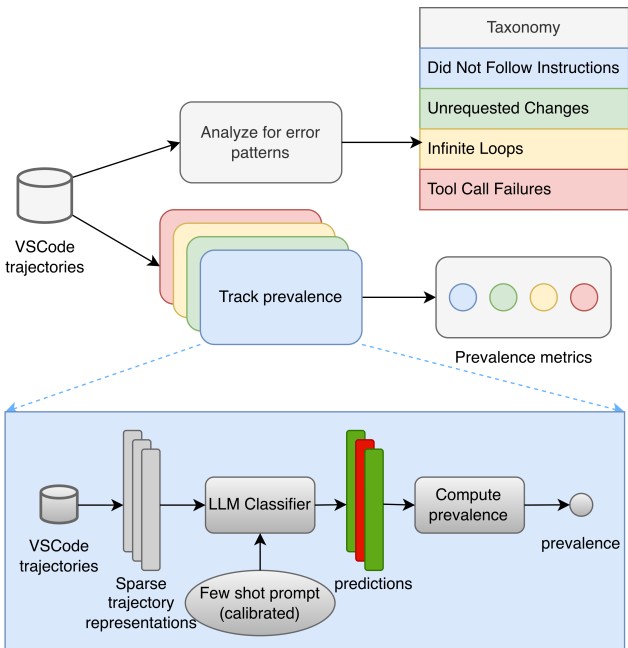

Figure 2: Overview of the misbehavior prevalence metrics computation: Taxonomy creation and trajectory classification

classifiers. We noted that models had different strengths allowing flexibility of use in various scenarios that require high precision versus high recall. As we intend to rely on the classifier's output to inform subsequent fixes, we set a high bar for precision (at least 80%). Claude Sonnet 4 had the best performance across all categories, with few-shot prompting (examples used real users' complaints sourced from internal feedback groups and in-conversation feedback forms).

Once we had validated the effectiveness of the classifier, we deployed all misbehavior classifiers on 10% of daily production traffic (8k trajectories) to establish the prevalence.

*2.2.2 Offline dataset* In addition to recurring runs, we constructed a static historical dataset by randomly sampling trajectories from five consecutive weeks of usage. The static dataset consists of 42,920 trajectories that contain real user sessions. The dataset serves as a baseline to understand historic trends in the misbehavior prevalence before we apply interventions.

*2.2.3 Misbehavior prevalence* On the static set, we observe that the Did Not Follow Instructions (Specification Drift) and Tool Call Failures had the highest average prevalence. Overall prevalence is around 29% (Table 1).

Note that the classifiers detect misbehaviors at trajectory-level so it is possible for a single trajectory to have multiple misbehaviors. At the same time, many of these behaviors are also mutually exclusive in nature i.e., ~65% of the trajectories where tool call failures are observed and ~45% of the trajectories where instruction following problems are observed are exclusive to those categories.

Table 2 shows the prevalence of different misbehavior categories for Claude Sonnet 4.5 and Claude Opus 4.5. Specification drift (DNF and UC) problems improved for Opus 4.5 but prevalence for infinite loops and tool call failures increased. All of the improvements or regressions are statistically significant (p-value: < 0.00001 for infinite loops and DNF, p-value: 0.00123 for UC) except for tool call failures (p-value: 0.878). This indicates that although newer models can improve some of the misbehaviors, not all of them are impacted significantly. Hence, other solution to address these problems are necessary.

## 3 Self intervention

As explained in Section 2, once we identify the classes of misbehaviors and their prevalence, we set out to implement a course-correction system that detects misbehaviors as they are happening, reflects on the agent's trajectories till the intervention point, and offers guidance to course correct.

The main coding agent follows a custom harness built on top of the ReACT pattern [24]. The agent takes a task description as an input and calls various tools as it progresses. Once the agent thinks the task is completed or reaches a termination condition, the agent stops by producing a diff. All the steps the agent has taken, its reasoning, tools calls, and interactions are recorded explicitly as a trajectory. At any given step $t$, the trajectory is

$$\text{Trajectory}_t = (u_1, a_1, acc_1, o_1, a_2, acc_2, o_2, ... u_t, a_t, acc_t, o_t) \quad (1)$$

Here, $u_i$ are the user messages, which typically provide the initial task specification. Sometimes, the users may subsequently provide more instructions with additional guidance or for accomplishing a followup task. $a_i$ are the assistant messages which include agent's reasoning and thoughts. $acc_t$ are the actions taken by the agent (tool calls) and $o_i$ are the observations made upon taking those actions.

The next assistant message and action pair $\langle a_{t+1}, acc_{t+1} \rangle$ is conditioned on the trajectory $\text{Trajectory}_t$ and new user input $u_{t+1}$ (if there is any), i.e.,

$$(a_{t+1}, acc_{t+1}) \sim f(\cdot \mid \text{Trajectory}_t, u_1, u_{t+1}, m) \quad (2)$$

| Misbehavior Category | Prevalence in Sonnet 4.5 (%) | Prevalence in Opus 4.5 (%) |
|---|---|---|
| Did Not Follow Instructions | 18.69 | 13.19 |
| Unrequested Changes | 6.55 | 5.2 |
| Infinite loops | 2.18 | 3.59 |
| Tool Call Failure | 10.99 | 11.08 |

Table 2: Prevalence of Misbehaviors in Sonnet 4.5 and Opus 4.5

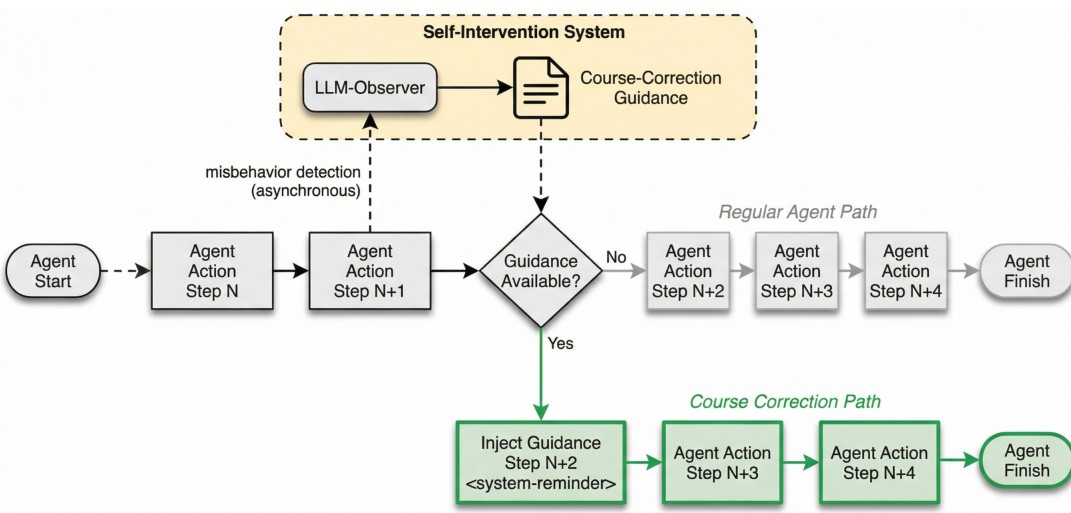

Figure 3: The Self-Intervention system architecture. The observer runs asynchronously to prevent latency regressions in the main SWE agent loop, injecting guidance via system-reminders only when results are available.

where $m$ is metadata such as tool descriptions, system prompts, etc. When $acc_{t+1}$ is executed, it yields $o_{t+1}$, which is appended to the trajectory ($Trajectory_{t+1}$).

## 3.1 Reflection and Guidance Generation

When the agent is working on a task, at fixed intervals, we make an async call to invoke our misbehavior detection system. This inspects the recorded trajectory till that point and returns feedback.

At any given step $k$, the trajectory that encompasses all the user messages, assistant messages, actions, and observations is represented as $Trajectory_k$ (as described in equation (1)). The misbehavior detection system ($MB_k$) then consumes $Trajectory_k$ and the taxonomy of misbehaviors ($\Gamma$), explained in Section 2, to produce feedback.

$$Feedback_k = MB_k(Trajectory_k, \Gamma) \qquad (3)$$

The contents of the feedback are returned as a tuple with two items: (1) A binary output indicating whether $Trajectory_k$ has any misbehaviors manifested in it, and (2) The name(s) of the misbehavior class(es) identified which may include Specification Drift, Reasoning Problems, or Tool Call Failures.

## 3.2 Course Correction

Feedback from the misbehavior detection system informs the course-correction actions. Guidance is dynamically generated via specific instructions that we pass in the prompt (from a store) and it is in plain text, composed of various DOs and DONTs. Usually, it nudges the model to self-reflect and take an alternative action or path. Finally, the guidance $Guidance_t(k)$ is appended to the recorded trajectory and passed as input ($AgentInput_{k+1}$) to the agent again to generate next set of thought, action pairs ($\langle a_{t+1}, acc_{t+1} \rangle$).

$$AgentInput_{k+1} = Trajectory_k + Guidance_t k \qquad (4)$$

## 3.3 System architecture

When designing the self-intervention system, our goal is to ensure that its LLM-based classifier and intervention mechanism integrate seamlessly into the agentic harness without negatively impacting user experience, whether through regressions or increased latency. Latency is especially important since users expect prompt responses and actions from the agent. To achieve this, we integrated the observer so that it never blocks the main agentic harness, invoking it asynchronously at a fixed interval of $k$ steps.

At every step in the agent loop, specifically after an agent action, we check if a result from the asynchronous observer is available. If it is, then the response is parsed to extract the misbehavior category (if any), the reasoning and, any course correction instructions. The instructions are then injected within special XML tags called system-reminder into the agent conversation. These interventions are not observable to the end user; they only influence the agent's behavior.

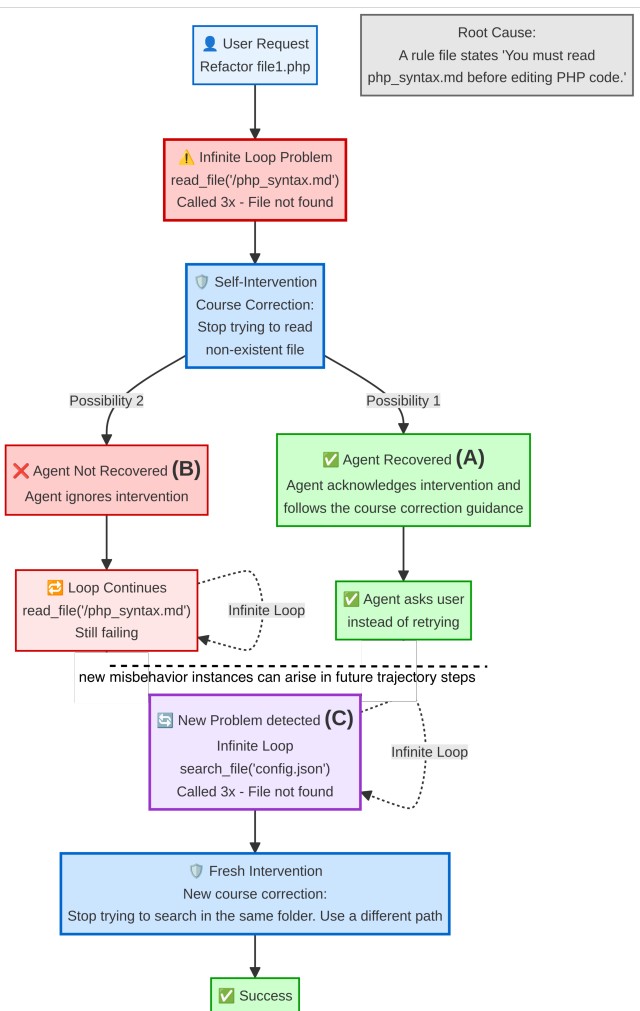

Figure 4: Self-Intervention mechanism addressing agent infinite loops misbehavior. The diagram illustrates recovery (A) and non-recovery (B) trajectories following an intervention. Recovery occurs if the specific misbehavior (e.g., redundant calls to read_file(php_syntax.md)) does not recur. Conversely, non-recovery is defined by the reoccurrence of the same pattern of actions constituting that misbehavior. Different instances of agent misbehavior (C) can arise later, which would trigger new, separate intervention steps.

## 4 Experiment setup

We designed our experiments to address the following main research questions:

(1) RQ 1: Does self-intervention effectively resolve agent misbehaviors?
(2) RQ 2: Does self-intervention reduce the magnitude of agent misbehaviors?
(3) RQ 3: What is the impact of self-intervention on our standard, pre-established collection of agent metrics?
(4) RQ 4: Under what circumstances do agents recover from failures, and when do they not?

To answer some of these questions, we ran a live A/B test [17] to understand the impact of self intervention, where we split the live production traffic into two groups (50-50) and let the experiment run for 15 days. In the treatment group, the self intervention system is enabled, whereas in the control group, it is disabled. We also curated distinct datasets tailored to each question and proposed relevant evaluation metrics when appropriate.

### 4.1 RQ1: Does self-intervention effectively resolve agent misbehaviors?

To answer this question, we collected 10,554 agent trajectories from the A/B test treatment group specifically where interventions were triggered, encompassing all misbehavior classes. Note that while the self-intervention system is enabled for the entire treatment group, interventions are only applied in trajectories when misbehaviors are detected. We categorized each trajectory based on the number of self-interventions: those with a single intervention and those with multiple interventions. This distinction enables targeted analysis of self-intervention effectiveness across failure types.

We use LLM-as-judge technique [27] to classify trajectories as recovered and non-recovered. We pass all steps from the trajectory before intervention happened, the intervention step, specific misbehavior detected with reasoning, course-correction guidance, and 15 steps after the intervention happened as inputs to the judging LLM. The judge marks an agent trajectory as 'recovered' if the agent no longer exhibits the detected misbehavior and makes clear forward progress in the post-intervention steps.

To provide an example, if the agent is repeatedly reading an unchanged file with a tool call ($read\_file(file1.cpp)$), the Self Intervention system provides guidance to stop doing that and asks the agent to reuse existing file content from its history. We consider the agent as recovered if we do not see the same tool call with the same argument again ($read\_file\ (file1.cpp)$). We manually verified tens of trajectories classified by the LLM judge to understand its quality. We found its precision to be 85.71%, which indicates that the LLM judge is good at not marking not-recovered trajectories as recovered.

We measure Recovery Rate in the trajectories to quantify the number of times the agent was able to recover successfully from an instance of a misbehavior and did not manifest that specific misbehavior again in the post-intervention steps.

$$Recovery\ Rate = \frac{Number\ of\ recovered\ misbehaviors\ \times 100}{Total\ number\ of\ observed\ misbehaviors} \quad (5)$$

Tables 3 and 4 present the agent's recovery rates from different misbehavior categories. Trajectories where the intervening agent made a single self intervention (Table 3), the agent demonstrates a high overall recovery rate of 90.93%.

In trajectories where multiple interventions were made by the agent (Table 4), the overall recovery rate drops to 79.07%, suggesting these are complex tasks with many user and agent turns.

| Misbehavior Category | Sample Size | Agent Recovered | Agent Not Recovered | % Agent Recovery Rate |
|---|---|---|---|---|
| Reasoning Problems | 963 | 908 | 55 | 94.29 |
| Tool Call Failure | 2604 | 2386 | 218 | 91.63 |
| Specification Drift | 1627 | 1429 | 198 | 87.83 |
| Overall | 5194 | 4723 | 471 | 90.93 |

Table 3: Recovery rate for single-intervention conversations, as determined by the LLM judge. This estimate is conservative: if the evidence is insufficient to verify recovery, the judge labels the post-intervention trajectory as "not recovered."

| Misbehavior Category | Sample Size | Agent Recovered | Agent Not Recovered | % Agent Recovery Rate |
|---|---|---|---|---|
| Reasoning Problems | 2414 | 1781 | 633 | 73.78 |
| Tool Call Failure | 1531 | 1346 | 185 | 87.92 |
| Specification Drift | 1415 | 1111 | 304 | 78.52 |
| Overall | 5360 | 4238 | 1122 | 79.07 |

Table 4: Recovery rate for conversations with multiple-interventions, as determined by the LLM judge. This estimate is conservative: if the evidence is insufficient to verify recovery, the judge labels the post-intervention trajectory as "not recovered."

## 4.2 RQ2: Does self-intervention reduce the magnitude of agent misbehaviors?

To answer this question, we conducted a "shadow mode" experiment (separate from the A/B test described earlier) i.e., we operationalized the self-intervention system on a fraction of the live traffic. For a random subset, we only ran the misbehavior detection component of the system (without intervention). That way we ended up collecting 3864 invocations (711 trajectories) with just misbehavior detection and 4168 invocations (759 trajectories) with misbehavior detection *plus* applying course-correction, upon flagging a misbehavior. The invocation interval $k$ was set to 5 steps.

Then, we calculated Misbehavior Rate (MR), which is the percentage of misbehaviors, detected by the misbehavior detection part of the system, over the total number of invocations. MR for the subset where course-correction is applied is 15.14% whereas for the subset with just detection enabled is 18.61%. The difference is statistically significant based on a two-proportion z-test at a confidence interval of 99.9% ($p-value = 0.00003274$). This indicates that Self Intervention with course-correction is introducing positive behavior changes and steering the agent in the right direction.

## 4.3 RQ3: What is the impact of self-intervention on our standard, pre-established collection of agent metrics?

From the live A/B test, we observe the treatment group demonstrated a reduction in the overall tool call failure rate metric i.e., percentage of failed tool calls across all the trajectories. The treatment group recorded 5.07% tool call failure rate, which is a 4.2% reduction from the control group (5.29%). The difference is statistically significant (99% CI, $p-value = 0.0096$).

To measure the impact of the intervention system on agent's effectiveness with respect to accomplishing coding tasks i.e., the agent's ability to accomplish tasks faster and with fewer resources in a given agentic session, we measured three session—level metrics.

We observed that Token Usage per Session decreased by 5.3% (95% CI, $p-value = 0.003$), indicating more efficient resource utilization. More notably, Engineer Interventions per Session—a direct measure of human oversight required, decreased by 4.2% (95% CI, $p-value = 0.014$), suggesting the self-intervention mechanism enables the agent to operate more autonomously with reduced need for human correction. We also observed directional improvements in agent execution time per session (-4.3%, $p-value = 0.073$), though this did not reach statistical significance at the $\alpha = 0.05$ level.

| Metric | $\Delta\%$ |
|---|---|
| Tool Call Failures | $-4.2\%^{**}$ |
| Tokens per Session | $-5.3\%^{**}$ |
| Engineer Interventions per Session | $-4.2\%^{*}$ |
| Agent Execution Time per Session | $-4.3\%$ |

$^{**}p < 0.01$, $^{*}p < 0.05$

Table 5: Results from the live A/B test

With this result we conclude that course-correction has been effective when it comes to reducing Tool Call Failure misbehaviors, as well as reducing both computational costs and the burden on human engineers to steer the agent towards task completion.

## 4.4 RQ4: When do agents recover from failures and when do they not?

We performed qualitative data analysis [1] on a randomly selected subset of recovered and non-recovered trajectories.

We performed coding [22] manually with four members of the team annotating the LLM-classified trajectories. Coding is the process of labeling and organizing qualitative data to identify different themes and the relationships between them. The labeling process involved three steps: 1. The team got together and came up with four categories (labels) for the recovered trajectories and five for the non-recovered trajectories to represent the scenarios. First, we

identified some categories that are intuitive and refined them as we sift through the trajectories to arrive at a final set of categories. 2. Four members of the team performed labeling independently. 3. The team got together again to reconcile their understanding and made modifications to the labels as necessary.

The classes and their descriptions are provided in Table 6. We coded trajectories (both recovered and non-recovered) to fit into each of these classes based on their recovery status. Then, we calculate and report the percentage of trajectories that fall into each of the classes.

| Outcome | Reason | % |
|---|---|---|
| Recovered | Recover from infinite loops | 39 |
| | Remind original task with updated plan | 26 |
| | Ask agent to not overdo | 17 |
| | Provide right tool arguments | 17 |
| Not Recovered | Agent ignored the course-correction instructions | 37 |
| | Premature termination | 22 |
| | Mechanical failures (IDE, Tools, etc) | 19 |
| | Hard merge conflicts | 11 |
| | False Negatives | 11 |

Table 6: Qualitative coding categories by recovery outcome

The distribution of intervention types across recovery outcomes can be seen in Table 6. Agents most frequently recovered from infinite loops (39%). Reminding agents of the original task with an updated plan proved effective in 26% of cases. Additionally, instructing agents to avoid scope creep (17%) and providing correct tool arguments (17%) successfully redirected agent behavior toward task completion.

Conversely, agents failed to recover frequently because of their disregard to the course-correction instructions (37%). Other failure modes included premature termination of the task by the agent (22%), mechanical failures in IDEs or the tools that the agent calls (19%), complex merge conflicts that are hard to fix (11%), and false negatives caused by the agents incorrectly assessing their state (11%).

Figure 5 illustrates a representative recovery scenario for tool call failures. First, the agent calls a tool (`bash`) with incorrect arguments. Then, the intervention mechanism detects error and prescribes the right command i.e., to run `activate.sh` before executing tests, which helps the tool call to complete execution successfully.

## 5 Threats to validity

### 5.1 Generalizability

The agentic misbehavior detection and self intervention mechanism presented in this paper has only been tested and deployed at CompanyA. Hence, there is a potential threat of drawing any general conclusions from the result of the analysis. The improvement achieved in VSCode trajectories using the self intervention technique might not hold elsewhere. Similarly, the taxonomy that we defined for misbehavior categories might not translate well into other platforms since they were designed by analyzing CompanyA-specific trajectories.

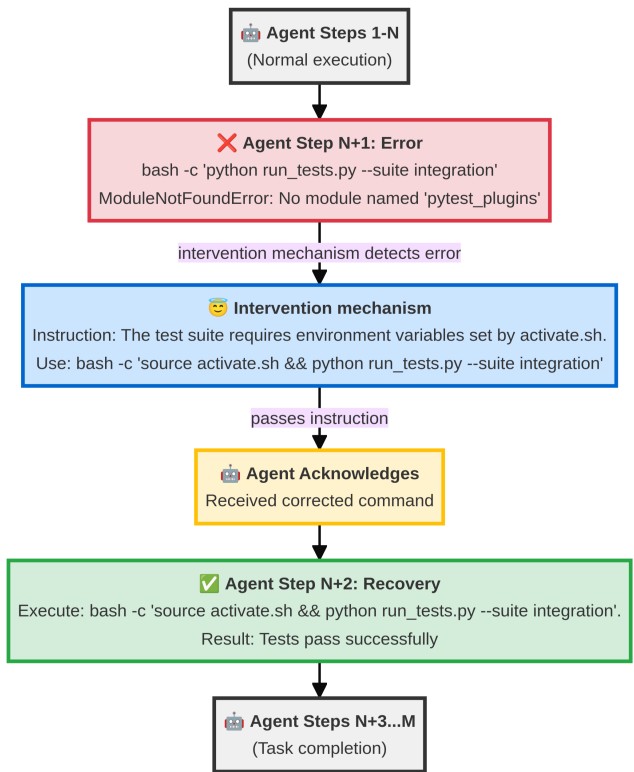

Figure 5: Self-intervention mechanism showing a tool call failure followed by intervention and recovery.

### 5.2 Internal Validity

We have taken appropriate measures to minimize bias in the results presented in Section 4, including analyzing prevalence through weekly breakdowns and conducting A/B tests. Nevertheless, it is important to note that the complete elimination of bias cannot be guaranteed. The time period chosen for the experiment could involve other experiments which are beyond our control and could have influenced the observed prevalence metrics.

The taxonomy we developed does not encompass all possible forms of agentic misbehavior; rather, it focuses on categories that appeared most prevalent. We built classifiers only for these categories. To further validate our approach, we conducted a comprehensive analysis using an LLM-based judge to cross-check the selected categories. However, the impact of the self-intervention mechanism on misbehavior categories not included in our taxonomy remains unknown.

## 6 Related work

### 6.1 LLM-based Agents in Software Engineering

LLM-based assistance in software engineering has evolved from probabilistic code completion to autonomous, tool-augmented agents capable of planning, acting, and reflecting over multi-step tasks [11, 15, 16, 25]. Contemporary systems extend beyond code synthesis to program comprehension, test generation, and automated program

repair (APR), integrating capabilities such as repository navigation, build/test orchestration, and IDE instrumentation. Notable exemplars include RepairAgent [4], AutoCodeRover [26], and Open-Hands [21] for bug fixing, and multi-agent frameworks such as Chat-Dev [19] and MetaGPT [14] for collaborative software workflows. Despite broad industrial adoption, the internal decision-making dynamics—spanning planning, tool selection, and error recovery—and their failure modes in production remain insufficiently characterized. In contrast to these surveys and systems that evaluate agent capabilities on curated benchmarks, our work provides the first large-scale empirical analysis of agentic misbehaviors observed in an industrial production environment with thousands of active users and millions of monthly conversational sessions.

## 6.2 Taxonomies of Agent Failures and Misbehaviors

Understanding agent misbehavior requires a principled taxonomy of error classes grounded in observed trajectories. Prior work spans: (i) theory-oriented taxonomies for general agents that distinguish reasoning, execution, and planning errors [28]; (ii) empirically derived classifications from open-source coding agents' GitHub issues [10]; (iii) multi-agent failure frameworks (e.g., MAST [6]) emphasizing system design, inter-agent coordination, and verification; and (iv) automated diagnostic frameworks such as AgentRx [3], which derives a cross-domain failure taxonomy from grounded theory and pinpoints critical failure steps in execution trajectories. While these schemata clarify failure loci (e.g., goal mis-specification, tool misuse, verification gaps), they under-specify phenomena endemic to industrial SE contexts—proprietary toolchains, heterogeneous legacy codebases, and organization-specific workflows. Our contribution complements these lines by deriving a taxonomy grounded in real developer interactions with more than a thousand MCP tools within a proprietary enterprise ecosystem, capturing failure modes specific to large-scale industrial deployments that are invisible in public datasets.

## 6.3 Analysis of Agent Behavior and Trajectories

Empirical study of agent behavior leverages execution trajectories—interleavings of model reasoning, actions (tool invocations), and observations (tool outputs)—as in ReAct-style agents [5, 18]. Recent methods include sequential pattern mining to surface behavioral motifs and anti-patterns [5], semantic coherence assessment linking intermediate reasoning to task goals, and LLM-as-judge pipelines for scalable annotation [13]; these are complemented by progress metrics, visualization, and debugger-like inspection tools. Building on this methodology, we instrument production systems to capture high-fidelity traces, then combine quantitative analyses of action patterns (e.g., oscillatory tool use, premature termination) with qualitative evaluations of reasoning coherence. LLM-based classifiers are calibrated on human-annotated samples to ensure reliability. Unlike prior trajectory studies that focus on traces from benchmark suites, we conduct analysis at the production scale, enabling statistically robust measurement of misbehavior prevalence and longitudinal trends that are infeasible with smaller datasets.

## 6.4 Runtime Intervention and Self-Correction Mechanisms

A complementary line of work mitigates agent misbehavior at inference time via runtime intervention rather than post-hoc analysis. Process Reward Models (PRMs) score intermediate steps to detect trajectory-level errors: SWE-PRM [12] deploys an inference-time PRM grounded in a taxonomy of common inefficiencies (redundant exploration, tool-use loops, failure to terminate), AgentPRM [23] jointly models short-term progress and long-term promise, and Choudhury [7] outlines a practical Monte Carlo rollout framework for scalable PRM training. Beyond reward modeling, runtime enforcement constrains agent behavior using explicit specifications: AgentSpec [20] introduces a lightweight DSL for runtime rules with triggers and enforcement actions, VIGIL [8] offers a reflective runtime for structured diagnosis and recovery, and ARM [2] demonstrates closed-loop remediation where agents monitor SLO-aligned indicators and execute corrective actions. Together, these mechanisms span a spectrum from soft guidance (reward signals) to hard constraints (policy enforcement). While these intervention approaches are designed and evaluated on controlled benchmarks, our work provides empirical evidence of misbehavior patterns at production scale, offering actionable insights for designing intervention strategies tailored to the specific failure modes encountered in real-world industrial settings.

## 7 Conclusion

In this paper, we introduced a taxonomy of agentic misbehaviors derived from a large-scale analysis of trajectories generated by AI agents when working on real-world software engineering tasks, and we presented a novel self-intervention system designed to automatically recover from these misbehaviors. Our findings show that automated self-intervention is a viable and effective strategy for improving the efficacy and reliability of software engineering agents. At the same time, we see several promising directions for future improvements. We noticed that in some cases when self-intervention fires with a delay, the agent was able to recover without the course correction guidance, making guidance redundant. Additionally, in cases of specification drift, course correction can nudge the agent to end the current turn and seek user input before proceeding. This can lead to an increase in user turns which is not ideal, although the interventions were justified to correct the drift. These observations highlight the importance of developing sophisticated intervention strategies for timely and effective guidance. Moreover, the difficulty in recovering from complex, multi-turn misbehaviors points to an opportunity to develop hierarchical intervention mechanisms, where the system can escalate from simple nudges to more comprehensive plan revisions. We believe, as agentic systems become more capable and autonomous, the ability to self-correct will become not just a desirable feature, but a requirement for their effective deployment.

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
