# OpenReview forum: "Recovering from Misbehaviors in Coding Agents"
_ACM.org/AIWare/2026/Conference — AIware 2026_

### Official Review · Reviewer_5xqd · 2026-03-03

**Rating:** 2
**Confidence:** 4

**Review:**

Strengths:
1. The paper is well-written and clearly structured, easy to follow.
2. The topic is interesting and practically valuable — recovering from agent misbehaviors at runtime is an important problem as coding agents become more widely adopted.
3. The misbehavior taxonomy is derived from real user feedback (disliked trajectories), which gives it practical relevance.

Weakness:
1. The paper only demonstrates through a production A/B test that "with intervention" is better than "without intervention", but does not show whether the proposed lightweight asynchronous system truly outperforms existing classic baselines such as Reflexion, LATS, and multi-agent debate-based correction mechanisms in terms of correction success rate and execution efficiency.
2. Introducing a continuously running intervener inevitably adds extra token consumption (including input context and generated judgment text). Although Table 5 shows an overall 5.3% reduction in tokens per session, the paper does not separately report the token overhead of the intervention system itself. A fine-grained cost-benefit analysis that breaks down the intervener's own cost versus the savings from reduced misbehaviors is missing.
3. Although the paper mentions manual inspection, it focuses mainly on precision. For false negatives (the agent misbehaves but the system fails to detect it, or the intervention fails but the LLM judge marks it as recovered), the paper lacks systematic evaluation. Moreover, the paper uses a fixed window (15 steps) to evaluate recovery, but in real complex coding tasks, errors often have delayed effects. This confirmation bias may lead to overestimation of the success rate (e.g., the reported 90%).
4. The asynchronous monitoring system checks trajectories every k steps. However, the paper does not sufficiently justify the choice of k. The paper should provide an ablation study on k, showing the trade-offs among detection timeliness, intervention success rate, and token consumption under different k values.
5. The definition of recovery is too loose and disconnected from task success. The paper defines recovery as "the agent no longer exhibits the detected misbehavior" (i.e., not repeating the same error). However, stopping one specific error does not mean the task is completed successfully — the agent may continue to fail in a different way. The paper does not report any end-to-end task success rate, which makes the 90% recovery rate less meaningful.
6. The reliability verification of the LLM-as-Judge is insufficient. The paper only states that the team "manually verified tens of trajectories", without reporting the exact number of verified samples, nor providing recall or inter-annotator agreement metrics. For an evaluation framework that draws core conclusions from 10,000+ trajectories, this level of verification is far from adequate, casting doubt on the credibility of the reported 90% recovery rate.

Questions:
1. Can you provide a quantitative comparison between this system and existing classic self-reflection frameworks in terms of success rate, total time, and token overhead?
2. For the 10% of failures that cannot be resolved by a single intervention, how does the system currently perform? Can you provide case studies?
3. As the base model changes, will the distribution of the three misbehavior categories change significantly? How well does the intervention system generalize to different base models?
4. On average, how much additional token overhead does the system introduce per trajectory? In a real large-scale high-concurrency production environment, could this surge in extra LLM calls easily hit API rate limits?

**Summary:**

This paper proposes an asynchronous self-intervention system to detect and correct misbehaviors of LLM-based agents in a large-scale production environment. The authors build a three-category taxonomy based on real production traffic and propose a monitoring and intervention mechanism for targeted correction. Evaluation on over 10,000 real-world trajectories and a production A/B test show high recovery rates (about 91% for single interventions and about 80% for multiple interventions), as well as significant reductions in tool call failures per session, token usage per session, and engineer interventions per session.

---

> ### Author Response · Authors · 2026-03-20
>
> We sincerely thank the reviewer for their valuable suggestions and constructive feedback. We address the mentioned questions below:
>
> ### Response to Question 1
> Addressed in this [comment](https://openreview.net/forum?id=gJ9pQ8xLs0&noteId=tJSscrG29Z) under the **Response to Weakness 4**.
>
> ### Response to Question 2
> For the ~10% of failures not immediately resolved by a single intervention, we observed a highly diverse range of trajectory outcomes. It is important to note that a failure to auto-recover from a specific misbehavior does not necessarily mean the overall engineering task failed. As detailed in our qualitative analysis in Table 6, the most frequent reason an intervention fails is that the agent simply ignores the injected course-correction instructions (37% of non-recovered cases). However, a failed intervention does not necessarily mean a failed task. When we look at the subsequent steps in these specific trajectories, we observe a diverse range of ultimate outcomes:
> - When the agent stubbornly repeats its error despite the system's reminder, the human user often manually intervenes in the chat to explicitly steer the agent back onto the correct path, leading to ultimate task success.
> - We also observed trajectories where the agent ignores the explicit system instruction in the immediate next step, but eventually naturally course-corrects and recovers on its own in subsequent steps.
>
> In other cases, the intervention is classified as "Not Recovered" simply because the opportunity for recovery is cut short, such as when the agent terminates the task prematurely before any meaningful forward progress can manifest
>
> ### Response to Question 3
> Yes, the distribution of misbehaviors changes significantly depending on the base model. When comparing Claude Sonnet 4.5 to Opus 4.5, as presented in the paper (Table 2), we found:
> - **Improvements**: Specification Drift issues decreased significantly (e.g., "Did Not Follow Instructions" dropped from 18.69% to 13.19%).
> - **Regressions**: Conversely, "Infinite Loops" increased from 2.18% to 3.59%.
>
> This demonstrates that while newer models mitigate some failure modes, they do not resolve all of them and can even slightly exacerbate others. Therefore, an automated intervention system remains necessary.
>
> **Generalizability of the Intervention System**
>
> The Self-Intervention system is model-agnostic and generalizes easily to different base models.
> The course-correction guidance is dynamically generated as plain text, consisting of straightforward DOs and DONTs.
> This guidance is seamlessly injected into the agent's context using standard `<system-reminder>` XML tags.
> Because our approach steers the agent using in-context natural language rather than relying on model-specific fine-tuning or proprietary architectural hooks, it is effective with any instruction-following base model.
>
> ### Response to Question 4
> Thank you for raising this practical concern. Here's the data:
> 1. **Token Overhead:** To manage context bloat, we optimized LLM observer calls, truncating tool outputs from trajectory that exceed 800 tokens. Per trajectory, the observer adds on average 6.5 calls, 177,450 input tokens and 2,060 output tokens. Compared to main orchestrator (3.3M input, 16K output across 34.4 calls per trajectory), this adds roughly 5.36% input and 12.5% output volume. However, as demonstrated in our live A/B test, early interventions offset this overhead to some extent, yielding a net 5.3% reduction in token usage per session.
> 2. **API Rate Limits:** Because the observer runs asynchronously every $k$ steps, request volume is distributed rather than spiked. We validated this in our 15-day live A/B test, successfully handling 50% of real production traffic for thousands of developers without hitting provider rate limits or bottlenecking infrastructure.
>
> ### Weakness 4
> > the paper does not sufficiently justify the choice of k.
>
> The invocation interval was set to $k=5$ . While a formal live ablation study was too risky for production traffic, our selection was strictly guided by an offline analysis of historical trajectories to balance cost, timeliness, and success rate. Offline analysis showed a median session length of 14 steps (average 34 steps).
> - **Token Cost** (Risk of low $k$): Misbehaviors rarely manifest immediately. Setting $k$ too low (e.g., 1–2 steps) triggers premature, redundant checks that drive up token consumption without proportional benefits and increase the risk of hitting API rate limits.
> - **Timeliness** (Risk of high $k$): Setting $k$ too high (e.g., 10+ steps) misses the critical window where a misbehavior originates. Delayed detection allows the agent to drift, making recovery significantly harder.
>
> Given the median length of 14 steps, $k=5$ places asynchronous checks at roughly the one-third and two-thirds marks. This optimally conserves resources during early execution while ensuring we catch and correct misbehaviors before they compound.

---

### Official Review · Reviewer_dkNV · 2026-03-08

**Rating:** 3
**Confidence:** 4

**Review:**

# Strength
+ The observation is practical and asynchronous
+ The evaluation is comprehensive and sound
+ Easy to read and follow

# Weakness
+ Heavy reliance on LLM-based classifiers for both prevalence estimation and intervention triggers without comprehensive ground-truth validation
+ The LLM-as-judge recovery assessment is supported by manually verified samples with 85.71% precision, but recall is unreported
+ The A/B test reports aggregate deltas but lacks stratified analyses
+ While related work is cited, there is no empirical comparison or discussion of trade-offs versus process reward models, exemplar-retrieval repair systems, or specified trajactory verifiers

**Summary:**

The paper studies misbehaviors of production coding agents and proposes a lightweight, asynchronous self-intervention system that detects trajectory-level failures and injects targeted guidance to nudge agents back on track. Using LLM-based monitors calibrated for three prevalence-grounded categories, the system operates without blocking the main agent and delivers “system-reminder” interventions. Evaluations over >10k trajectories with triggered interventions, a shadow-mode detection-versus-intervention study, and a 15-day 50/50 production A/B test show high LLM-judged recovery rates (≈91% for single interventions, ≈79% for multi-intervention episodes) and statistically significant reductions in tool call failures, token usage, and human engineer interventions per session.

---

> ### Author Response · Authors · 2026-03-20
>
> We sincerely thank the reviewer for their valuable suggestions and constructive feedback. We address the mentioned weaknesses below:
>
> ### Response to Weakness 1
> We sampled trajectories that received developer feedback through in-product buttons (‘like’ or ’dislike’) and feedback form on our internal coding agent. In total, we reviewed and labeled 328 trajectories: 247 exhibited at least one misbehavior, and 81 exhibited none. The dataset was partitioned among four annotators. Each annotator labeled their assigned trajectories as either (a) exhibiting misbehavior along with the specific misbehavior category, or (b) exhibiting no misbehavior.
>
> Regarding inter-annotator agreement, given the complex and nuanced nature of agent trajectories, we opted for a qualitative consensus-based labeling approach rather than independent, isolated parallel labeling. Annotators did not make final labeling decisions in isolation; instead, ambiguous cases and labeling rationales were discussed collectively to reach an informal consensus. Also this design choice reflects the provenance of our data: the like/dislike signals were produced by internal developers using the coding agent in practice, and “dislike” typically corresponded to concrete failures relative to established internal best practices. We treated this feedback as a high-quality filter for identifying problematic trajectories, and used the four annotators (who were also developers) to further bucket disliked trajectories into a structured set of misbehavior categories.
>
> ### Response to Weakness 2
>
> We manually reviewed a total of 51 trajectories to assess the judge's performance. We excluded 4 trajectories due to edge-case timing artifacts (e.g., premature or delayed interventions), leaving a refined dataset of 47:
>
> | Component | Count |
> | :--- | :--- |
> | TP | 30 |
> | FP | 5 |
> | FN | 3 |
> | TN | 9 |
>
> Based on these manually verified labels, the judge achieved a precision of 85.71% and 90.9% recall. We will add a concise summary of these evaluation details to Section 4.1.
>
> ### Response to Weakness 3
>
> We completely agree that a stratified analysis, such as breaking down the A/B test results by task complexity, programming language, etc would provide a much deeper understanding of where the self-intervention system is most effective.
> However, our live A/B test was specifically designed and instrumented to evaluate high-level, aggregate online metrics, such as Tool Call Failures, Token Usage, and Engineer Interventions per Session. At present, our production telemetry does not natively link these session-level experiment logs with granular task metadata. Consequently, retroactively extracting, joining, and validating this data across millions of production events to perform a reliable post-hoc stratified analysis is prohibitively difficult given our current infrastructure constraints.
>
> ### Response to Weakness 4
> We agree that positioning our work against exemplar-retrieval systems, PRMs, classic self-reflection frameworks and trajectory verifiers is valuable. In the camera-ready version, we will strengthen Section 6 to more clearly situate our approach. The key distinctions are:
>
> 1. **Synchronous blocking and latency overhead.** The cited baselines (Reflexion, LATS, multi-agent debate) operate synchronously, introducing unacceptable latency for an end-user waiting in a live IDE. For example, SWE-PRM blocks the agent every 5 steps, with monitoring calls costing approximately 6.2× more than the main model itself. Our system, by contrast, runs asynchronously and never blocks the main agent loop.
>
> 2. **Scalar signals vs. actionable guidance.** PRMs and trajectory verifiers produce scalar signals (scores, rankings, binary labels) suited for selection, reranking, or offline evaluation, but insufficient for live recovery: a scalar can flag an error but cannot specify *how* to fix it. Our LLM-Observer instead produces natural-language feedback injected mid-trajectory as a system reminder, directly telling the agent what went wrong and how to adjust.
>
> 3. **Exemplar-retrieval overhead.** Exemplar-retrieval repair systems require inserting concrete recovery examples, which has two drawbacks: (i) detailed exemplars increase prompt length and token cost, eroding real-time IDE performance; (ii) covering diverse coding misbehaviors demands a large, curated library. Our approach generates on-the-fly, context-specific interventions without storing or injecting exemplars.
>
> We will revise the related work section to articulate these distinctions, emphasizing that our contribution focuses on lightweight, textual, mid-trajectory self-intervention designed for a real-time production coding environment.

---

### Official Review · Reviewer_YmC5 · 2026-03-11

**Rating:** 3
**Confidence:** 4

**Review:**

Positives
+ The paper studies prevalent misbehaviors in real production coding agents and proposes a practical mechanism to detect and recover from such issues.
+ The approach is evaluated through a live A/B test in a production environment and demonstrates improvements in several system-level metrics.

Negatives
- The reliability of the reported results remains difficult to assess because key analyses rely on LLM-based classifiers and an LLM-as-judge, yet their evaluations are only partially reported.

Questions

1. The annotation process for the misbehavior taxonomy is unclear (line 172). The paper mentions that “hundreds of trajectories” were inspected, but it is unclear how many trajectories or misbehaviors were annotated. How many annotators were involved, and what was the inter-annotator agreement?

2. In Section 2.2, the performance of the LLM-based classifiers is not clearly reported. The paper mentions that precision exceeds 80%, but the exact precision values and, more importantly, the recall are unclear. Without these details, it is difficult to assess whether the reported misbehavior prevalence is reliable.

    In Section 4.1 (RQ1), the paper relies on an LLM-as-judge to classify trajectories as recovered or non-recovered (line 544). However, the validation of the judge is limited: only “tens of trajectories” were manually checked and the reported precision is 85.71%, while the recall is not reported. Without a clearer evaluation of the judge’s performance, the reported recovery rate may not be reliable.

3. The novelty of the proposed misbehavior taxonomy is somewhat unclear. Prior work has proposed several taxonomies for agent failures and misbehaviors. Could the authors clarify how the proposed taxonomy differs from or extends existing taxonomies?

**Summary:**

This paper studies the agent production trajectories and identifies three categories of agent misbehaviors: Specification Drift, Reasoning Problems, and Tool Call Failures. It develops an asynchronous monitoring system that detects misbehaviors from agent trajectories and injects guidance to steer the agent toward corrective behavior. Experimental results show strong recovery rates, with 90.93% recovery for trajectories requiring a single intervention and 79.07% for those requiring multiple interventions. A live A/B test in a production environment further demonstrates statistically significant reductions in production metrics such as Tool Call Failures, Tokens per Session, and Engineer Interventions per session.

---

> ### Author Response · Authors · 2026-03-20
>
> We sincerely thank the reviewer for their valuable suggestions and constructive feedback. Below, we address each of the specific questions.
>
> ### Response to Question 1
>
> We sampled trajectories that received developer feedback through in-product buttons (‘like’ or ’dislike’) and feedback form. In total, we reviewed and labeled 328 trajectories: 247 exhibited at least one misbehavior, and 81 exhibited none. The dataset was partitioned among four annotators. Each annotator labeled their assigned trajectories as either (a) exhibiting misbehavior along with the specific misbehavior category, or (b) exhibiting no misbehavior.
>
> Annotators conducted random cross-checks of one another’s labels. However, we did not obtain classical inter-annotator agreement by having multiple annotators independently label the same trajectories given the complex and nuanced nature of agent trajectories. This design choice reflects the provenance of our data: the like/dislike signals were produced by internal developers using the coding agent in practice, and “dislike” typically corresponded to concrete failures relative to established internal best practices. We therefore treated this feedback as a high-quality filter for identifying problematic trajectories, and used the four annotators (who were also developers) to further bucket disliked trajectories into a structured set of misbehavior categories.
>
> Although each trajectory was labeled by a single primary annotator, label decisions were not made in isolation. Annotators discussed ambiguous cases and their labeling rationale with one another prior to finalizing labels, yielding an informal consensus process that increased our confidence in label reliability.
>
> ### Response to Question 2
> **LLM-based classifiers performance:**
>
> We deliberately optimized our LLM-based classifiers to favor high precision (targeting >= 80%) over high recall. This was a conscious choice to ensure that any flagged trajectory genuinely exhibited the misbehavior, preventing our prevalence estimates from being artificially inflated by false positives. Because of this strict precision threshold, our recall is naturally lower. Consequently, the 29.2% overall misbehavior prevalence reported in Table 1 serves as a strict and reliable lower bound; the true occurrence of these misbehaviors in production is likely even higher.
>
> | Classifier | #samples | Precision % | Recall % | F1 score % |
> | :--- | :--- | :--- | :--- | :--- |
> | Loops | 50 | 78.9 | 71.4 | 75 |
> | DNF | 49 | 92.308 | 48 | 63.158 |
> | Unrequested changes | 40 | 83.333 | 25 | 38.462 |
> | Tool call failure | 55 | 87.5 | 58.3 | 70 |
>
> (Note: While we targeted a strict 80% precision minimum, the Loops classifier performed marginally below this threshold at ~79%, which we will note in the text).
> We will add the above evaluation data for our classifiers to Section 2.2.1 in the camera-ready version.
>
> **LLM-as-a-judge performance:**
>
> The use of strong LLMs as judges has been extensively validated as a reliable, scalable proxy for human evaluation, reaching human-level agreement. We manually reviewed a total of 51 trajectories to assess the judge's performance. Four trajectories were excluded from the final evaluation: one for incorrect misbehavior detection, two due to premature intervention at the final step preventing accurate post-intervention labeling, and one because the intervention was delayed by one step, allowing the agent to naturally recover.
> This left a refined dataset of 47 trajectories. The performance of the LLM-as-a-judge on this set is detailed below:
> | Component | Count |
> | :--- | :--- |
> | TP | 30 |
> | FP | 5 |
> | FN | 3 |
> | TN | 9 |
>
> Based on these manually verified labels, the judge achieved a precision of 85.71% (as noted in Section 4.1) and a strong recall of 90.9%. We will add a concise summary of these evaluation details to Section 4.1 in the camera-ready version.
>
> ### Response to Question 3
>
> We do not claim to propose a novel taxonomy but we do derive our misbehaviors from production to ensure relevance. We perform a bottom-up grounding of these failure modes by looking at production trajectories with negative developer feedback. We filtered the failure modes to the most frequently occurring misbehaviors (Did Not Follow Instructions, Infinite Loops) to ensure we are targeting real problems within the company. We then draw on high-level categorizations (Specification Drift, Reasoning Problems) established by prior work [[TRAIL](https://arxiv.org/abs/2505.08638), [When Agents Go Astray](https://arxiv.org/abs/2509.02360), [Understanding Code Agent Behaviour](https://arxiv.org/abs/2511.00197)] to structure these failure modes. The resulting taxonomy maps our production-sourced failure modes to the high-level categories from existing literature.

---

> > ### Comment · Reviewer_YmC5 · 2026-03-21
> >
> > Thank you for the detailed clarifications.
> >
> > The responses improve the clarity of the paper, especially regarding the annotation process and the evaluation of the classifiers and the LLM-as-a-judge.
> >
> > I appreciate the clarifications.